# Gluten Induces Subtle Histological Changes in Duodenal Mucosa of Patients with Non-Coeliac Gluten Sensitivity: A Multicentre Study

**DOI:** 10.3390/nu14122487

**Published:** 2022-06-15

**Authors:** Kamran Rostami, Arzu Ensari, Michael N. Marsh, Amitabh Srivastava, Vincenzo Villanacci, Antonio Carroccio, Hamid Asadzadeh Aghdaei, Julio C. Bai, Gabrio Bassotti, Gabriel Becheanu, Phoenix Bell, Camillo Di Bella, Anna Maria Bozzola, Moris Cadei, Giovanni Casella, Carlo Catassi, Carolina Ciacci, Delia Gabriela Apostol Ciobanu, Simon S. Cross, Mihai Danciu, Prasenjit Das, Rachele Del Sordo, Michael Drage, Luca Elli, Alessio Fasano, Ada Maria Florena, Nicola Fusco, James J. Going, Stefano Guandalini, Catherine E. Hagen, David T. S. Hayman, Sauid Ishaq, Hilary Jericho, Melanie Johncilla, Matt Johnson, Katri Kaukinen, Adam Levene, Sarah Liptrot, Laura Lu, Govind K. Makharia, Sherly Mathews, Giuseppe Mazzarella, Roxana Maxim, Khun La Win Myint, Hamid Mohaghegh-Shalmani, Afshin Moradi, Chris J. J. Mulder, Ronnie Ray, Chiara Ricci, Mohammad Rostami-Nejad, Anna Sapone, David S. Sanders, Juha Taavela, Umberto Volta, Marjorie Walker, Mohammad Derakhshan

**Affiliations:** 1Department of Gastroenterology, Palmerston North District Health Board (DHB), Palmerston North 4442, New Zealand; 2Gastroenterology Unit, Milton Keynes University Hospital, Milton Keynes MK6 5LD, UK; 3Department of Pathology, Ankara University Medical School, Ankara 06620, Turkey; ensariarzu@gmail.com; 4Oxford & Department of Gastroenterology, Wolfson College, University of Oxford, Luton & Dunstable University Hospital Trust, Bedfordshire LU4 0DZ, UK; mikemarshmd@uwclub.net; 5Memorial Sloan Kettering Cancer Center, New York, NY 10065, USA; srivasa2@mskcc.org; 6Spedali Civili, Institute of Pathology, 25100 Brescia, Italy; villanac@alice.it (V.V.); anna.bozzola@gmail.com (A.M.B.); moriscadei@gmail.com (M.C.); 7Palermo, Internal Medicine Unit, Department ‘PROMISE’, University of Palermo, “V. Cervello Hospital”, 90100 Palermo, Italy; acarroccio@hotmail.com; 8Gastroenterology and Liver Diseases Reseach Center, Research Institute for Gastroenterology and Liver Diseases, Shahid Beheshti University of Medical Sciences, Tehran 1985717411, Iran; hamid.assadzadeh@gmail.com (H.A.A.); hamidmohaghegh@gmail.com (H.M.-S.); m.rostamii@gmail.com (M.R.-N.); 9Department of Medicine, Dr. C. Bonorino Udaondo Gastroenterology Hospital, Buenos Aires 1264, Argentina; jbai@intramed.net; 10Department of Medicine, University of Perugia School of Medicine, 06129 Perugia, Italy; gabassot@tin.it; 11Department of Pathology, Carol Davila University of Medicine & Pharmacy, 050474 Bucharest, Romania; gbecheanu@yahoo.com; 12Gastrointestinal Pathology Center of Excellence University of Pittsburgh Medical Center, Pittsburgh, PA 15213, USA; bellpd@upmc.edu; 13Department of Pathology, University of Milan-Bicocca, 20900 Monza, Italy; camillo.dibella@asst-monza.it; 14Medical Deptartment, Desio Hospital, 20832 Desio, Italy; caselgio@tiscali.it; 15Deptartment of Pediatrics, Universita Politecnica delle Marche, 60123 Ancona, Italy; c.catassi@univpm.it; 16Dipartimento di Medicina, Chirurgia e Odontoiatria, Scuola Medica Salernitana, Università di Salerno Gastroenterologia, 84084 Salerno, Italy; cciacci@unisa.it; 17Pathology Department, Grigore T. Popa University of Medicine and Pharmacy, 700115 Iasi, Romania; delia.ciobanu@umfiasi.ro (D.G.A.C.); mihai.danciu@umfiasi.ro (M.D.); maxim_roxxana@yahoo.com (R.M.); 18Gastroenterology and Pathology, Sheffield Teaching Hospitals, Sheffield S10 2JF, UK; s.s.cross@sheffield.ac.uk (S.S.C.); david.sanders1@nhs.net (D.S.S.); 19Department of Pathology, All India Institute of Medical Sciences, New Delhi 110029, India; prasenaiims@gmail.com; 20Department of Experimental Medicine, Section of Anatomic Pathology and Histology, Medical School, University of Perugia, 06125 Perugia, Italy; rachele.delsordo@unipg.it; 21Department of Pathology and Laboratory Medicine, University of Rochester, Rochester, NY 14607, USA; michael_drage@urmc.rochester.edu; 22Center for Prevention and Diagnosis of Coeliac Disease, Gastroenterology and Endoscopy Unit, Fondazione IRCCS Ca’ Granda Ospedale Maggiore Policlinico, 20122 Milan, Italy; lucelli@yahoo.com; 23Massachusetts General Hospital, Center for Celiac Research & Treatment, Boston, MA 02114, USA; afasano@mgh.harvard.edu (A.F.); annasapone@yahoo.it (A.S.); 24Anatomic Pathology Unit, Department ‘PROMISE’, Palermo, University of Palermo, A.O.U. Policlinico “P. Giaccone”, 17491 Palermo, Italy; adamaria.florena@unipa.it; 25Division of Pathology, Fondazione IRCSS Ca’ Granda-Ospedale Maggiore Policlinico, University of Milan, 20832 Milan, Italy; nicola.fusco@ieo.it; 26Department of Pathology, Queen Elizabeth University Hospital, Glasgow G51 4TF, UK; james.going@glasgow.ac.uk (J.J.G.); sarah.liptrot@ggc.scot.nhs.uk (S.L.); khunlawin.myint@ggc.scot.nhs.uk (K.L.W.M.); mohammad.derakhshan@glasgow.ac.uk (M.D.); 27Department of Pediatrics, University of Chicago Medicine, Chicago, IL 60637, USA; sguandalini@peds.bsd.uchicago.edu (S.G.); hjericho@peds.bsd.uchicago.edu (H.J.); 28Department of Pathology and Laboratory Medicine, Mayo Clinic, Rochester, MN 55907, USA; hagen.catherine@mayo.edu; 29Molecular Epidemiology & Public Health Laboratory, School of Veterinary Science, Massey University, Palmerston North 4472, New Zealand; d.t.s.hayman@massey.ac.nz; 30Dudley Group NHS Foundation Trust, Birmingham City University, Birmingham B5 5JU, UK; sauid.ishaq@nhs.net; 31Weill Cornell Medicine, New York, NY 10075, USA; mej9041@med.cornell.edu; 32Luton Dunstable University Hospital, Gastroenterology, Luton LU4 0DZ, UK; matthew.johnson@ldh.nhs.uk (M.J.); adam.levene@ldh.nhs.uk (A.L.); ronnie.ray@ldh.nhs.uk (R.R.); 33Department of Internal Medicine, Tampere University Hospital, 33100 Tampere, Finland; katri.kaukinen@uta.fi; 34Faculty of Medicine and Health Technology, Tampere University, 33100 Tampere, Finland; 35Department of Histopathology, Royal Wolverhampton NHS Trust, Newcross Hospitals, Wolverhempton WV10 0QP, UK; laura.lu@nhs.net; 36Department of Gastroenterology, Human Nutrition, All India Institute of Medical Sciences, New Delhi 110029, India; govindmakharia@gmail.com; 37Haematology Unit, Milton Keynes University Hospital, Milton Keynes MK6 5LD, UK; Sherly.Mathews@mkuh.nhs.uk; 38Institute of Food Sciences-CNR, Immuno-Morphology, 83100 Avellino, Italy; giuseppe.mazzarella@isa.cnr.it; 39Department of Pathology, School of Medicine, Shahid Beheshti University of Medical Sciences, Tehran 1985717411, Iran; afshinmo2002@gmail.com; 40Department of Gastroenterology and Hepatology, Amsterdam University Medical Center, 1081 HV Amsterdam, The Netherlands; cjmdl1@gmail.com; 41Department of Experimental and Clinical Sciences, University of Brescia, 25100 Brescia, Italy; chiara.ricci@unibs.it; 42Central Finland Central Hospital, 33520 Jyväskylä, Finland; juha.taavela@tuni.fi; 43Department of Medical and Surgical Sciences, University of Bologna, 40126 Bologna, Italy; umberto.volta@unibo.it; 44Department of Anatomical Pathology, College of Health, Medicine and Wellbeing, University of Newcastle, Callaghan, NSW 2308, Australia; marjorie.walker@newcastle.edu.au; 45College of Medical, Veterinary & Life Sciences, University of Glasgow, Glasgow G12 8QQ, UK

**Keywords:** non-coeliac gluten sensitivity, histology, normal mucosa, coeliac disease

## Abstract

**Background:** Histological changes induced by gluten in the duodenal mucosa of patients with non-coeliac gluten sensitivity (NCGS) are poorly defined. **Objectives:** To evaluate the structural and inflammatory features of NCGS compared to controls and coeliac disease (CeD) with milder enteropathy (Marsh I-II). **Methods:** Well-oriented biopsies of 262 control cases with normal gastroscopy and histologic findings, 261 CeD, and 175 NCGS biopsies from 9 contributing countries were examined. Villus height (VH, in μm), crypt depth (CrD, in μm), villus-to-crypt ratios (VCR), IELs (intraepithelial lymphocytes/100 enterocytes), and other relevant histological, serologic, and demographic parameters were quantified. **Results:** The median VH in NCGS was significantly shorter (600, IQR: 400–705) than controls (900, IQR: 667–1112) (*p* < 0.001). NCGS patients with Marsh I-II had similar VH and VCR to CeD [465 µm (IQR: 390–620) vs. 427 µm (IQR: 348–569, *p* = 0·176)]. The VCR in NCGS with Marsh 0 was lower than controls (*p* < 0.001). The median IEL in NCGS with Marsh 0 was higher than controls (23.0 vs. 13.7, *p* < 0.001). To distinguish Marsh 0 NCGS from controls, an IEL cut-off of 14 showed 79% sensitivity and 55% specificity. IEL densities in Marsh I-II NCGS and CeD groups were similar. **Conclusion**: NCGS duodenal mucosa exhibits distinctive changes consistent with an intestinal response to luminal antigens, even at the Marsh 0 stage of villus architecture.

## 1. Introduction

Wheat, especially for its gluten component, is well-known to be a worldwide major food antigen. As such, its role has been implicated in several disorders, including celiac disease, dermatitis Herpetiformis, wheat allergy, gluten-induced ataxia and, more recently, in non-coeliac gluten sensitivity (NCGS). The expanding spectrum of gluten related disorders, from typical CeD to atypical CeD presentation, has now advanced into a new dimension with emerging recognition of the condition known as “non-coeliac gluten sensitivity” [1,2,3]. ‘Non-coeliac’ implies the absence of autoantibodies in the presence of gluten-induced symptoms in susceptible individuals. Non-coeliac has also been classified as gluten sensitivity with normal or nearly normal histology. However, possible microscopic and sub-microscopic [4] enteropathies that may overlap with the NCGS group have not, so far, been systematically evaluated. In addition, evidence to support the view that NCGS could exclusively present with normal histology or a milder enteropathy is lacking. 

Systemic immune activation is reported to damage the intestinal epithelium [5,6,7] in NCGS. The genetic and host immune responses are recognised by increased intraepithelial CD3(+) T cells and increased cytokines as a response to a gluten challenge [8]. 

The factors contributing to the variability of mucosal changes in CeD are insufficiently explored and there are no acceptable explanations in the current literature as to why the immune reactions involved in NCGS and some CeD patients [6] induce only mild or minimal mucosal changes. 

In NCGS, similar to CeD, gluten and other factors may cause alterations in villus morphology [9,10] and remodelling of intestinal mucosa [11,12,13]. Nevertheless, severe gluten-induced mucosal damages in antibody-negative patients [14,15] is not recognized to be a feature of NCGS despite similar clinical presentation and identical HLA class II haplotype frequency [16]. When the initial gluten-sensitive seronegative individuals were reported [17], NCGS was not recognised. Therefore, the severity of small bowel mucosal damages in NCGS is still contentious and has become even more complicated since histology is not even a mandatory part of the diagnostic work-up for NCGS [18]. Indeed, the exclusion of CeD is frequently assumed through negative serology. Erroneous reports of mucosal “atrophy” also preclude accurate understandings of the mucosal histopathogenesis. In fact, neither CeD nor NCGS mucosal changes could be due to an atrophic process simply because structural recovery of villi ensues in most cases following removal of the immunogenic antigens [19]. 

New data on NCGS histology presented in this report quantified the morphological changes in CeD with milder enteropathies (Marsh I-II) in comparison with NCGS, with the aim of delineating the histological spectrum of NCGS and its distinction from normal intestinal mucosa. In addition, we aimed to (i) quantify the eosinophils as an additional cause of NCGS and (ii) delineate clustering or linear distributions of IELs in villus epithelium or deeper lamina propria [20,21,22]. 

Furthermore, we assessed the double-blind placebo-controlled (DBPC) gluten challenge diagnostic strategy compared to an open gluten challenge policy for the diagnosis of NCGS.

## 2. Methods and Materials

This multicentre study was designed by clinicians and expert pathologists at the International Meeting on Digestive Pathology, Bucharest 2017. The study is based on a harmonised methodology for histological evaluation as conducted between investigators with interobserver agreement. Participants consisted of 23 tertiary centres with clinical and pathological expertise in CeD, NCGS, DH, and gluten or wheat allergies from 9 countries on 4 continents. Archived histology slides of 698 patients were randomly recruited from Australia (*n* = 20), Finland (*n* = 20), India (*n* = 25), Iran (*n* = 37), Italy (*n* = 239), Romania (*n* = 10), Turkey (*n* = 30), UK (*n* = 166), and USA (*n* = 151) (Appendix A). Those slides were from patients diagnosed with CeD Marsh I-II histology (*n* = 261), NCGS (*n* = 175), and 262 controls fulfilled our inclusion and exclusion criteria as formulized in a single agreed protocol. The inclusion and exclusion criteria for controls were similar to our previous study [9], as follows: The control group were included only if they had negative tTG and/or EMA. We included patients with anaemia and weight loss unrelated to the upper gastrointestinal tract with normal biopsies. Patients with infection (e.g., H pylori), bacterial overgrowth, functional dyspepsia, history of major abdominal surgery, liver disease, food intolerance, decompensated diabetes mellitus, or thyroid disease, patients using non-steroidal anti-inflammatory drugs (NSAIDs), and patients with any other liver or gastrointestinal disorders or diagnosis were excluded.

Patients were diagnosed in leading tertiary centres and biopsies were performed based on presentation with symptoms compatible with CeD or NCGS. All patients included in this study (CeD and NCGS) were untreated and the intestinal biopsies were taken before starting a gluten free diet. Details of inclusion and exclusion criteria are delineated in Appendix A.

Biopsies were taken both from the duodenal bulb (D1) and distal duodenum (D2) as recommended by the International Guidelines. All selected cases had concordant pathology in terms of Marsh I or II histology in D1 and D2 biopsies. However, histopathological measurements were limited to the well-oriented D2 biopsies as villi are shorter and architecturally distorted due to Brunner’s glands in the duodenal bulb. 

Our morphometric and quantitative examinations provided objective measurement of enteropathy by using villus height (VH), crypt depth (CrD), VD:CrD ratio and intraepithelial lymphocyte count (per 100 enterocytes). Intraepithelial lymphocytes were counted per 100 enterocytes covering the villous epithelium of the entire villus for the total IEL count of each case. IEls per individual villus were calculated by counting 10 consecutive well-oriented villi and then averaging, while IELs per individual crypt were calculated by counting 10 consecutive well-oriented crypts and obtaining the average. 

Some studies suggest a continuous and even distribution of IELs is indicative of CeD [23] and a non-specific increase in IELs is more likely to decrease along the villus tip. Therefore, we assessed the percentage of villi with increased IELs (PVIIEL) and also the percentage of villi with even distribution of IELs (PVEIEL) and their distribution in the crypt epithelial mucosa. 

Further evaluation was made of the inflammatory cell infiltrate in the lamina propria with a particular focus on eosinophil count (>5 eosinophils per high-power field (HPF) at ×40 magnification) [20,22,24]. A supplementary micrograph summarizes the steps included in the morphometric analysis of the duodenal mucosae.

Interclass correlation coefficient: see Appendix A.

### 2.1. Ethical Considerations

This study involved the rescoring of archived histology slides and the removal of all identifiable medical information, and analyses were performed using anonymized data. The data collection was in line with clinical best practice policies with approval by research and development departments of countries involved under the Reg.No 565 at Milton Keynes University Hospital, Milton Keynes, UK. The study was also approved by the ethical committee of the Research Institute for Gastroenterology and Liver Disease, Shahid Beheshti University of Medical Science, Tehran under the Ref.No: IR.SBMU.RETECH.REC.1399.1003.

### 2.2. Statistical Analysis

The distribution pattern of all continuous variables across the three main groups of NCGS, CeD and control biopsies are presented as median and interquartile range (IQR). Differences between NCGS and other groups were tested using both adjusted and non-adjusted models. In models, various general linear models were used, in which the histological parameters in a continuous form (or their log-transformed equivalent when data were obviously asymmetric, defined by skewness and kurtosis beyond the range of −1 to 1) were the dependent variables and histological groups were independent variables. Country, age, and sex were used as adjusting factors and covariates. For receiver operating curve (ROC) analyses of IEL density in the overall cohort and subgroups, area under the curve area (AUC) was calculated under non-parametric assumption. All tests of significance were set to *p* < 0.05 in two-sided tests.

## 3. Results

### 3.1. A. General Characteristics of Groups

In this multicentre cross-sectional study, endoscopic biopsy specimens from a total of 698 participants were evaluated, including 175 NCGS patients, 261 CeD patients, and 262 control subjects. Details of recruited subjects by group, centre, and country are presented in the Appendix A. The median age of the NCGS group (37·0, IQR: 27·0–47·0) was similar to the CeD patients (38·0, IQR: 23·0–51·0) (*p* = 0·839) but younger than the control group (43·0, IQR: 30·5–57·0), (*p* < 0·001). Only 4·0% of NCGS patients were under 18 years, similar to the control group (5·3%, *p* = 0·650) but significantly different from the CeD group, where 16.5% were under 18 years (*p* < 0·001). A total of 82·9% of NCGS patients were women, which was higher than the proportion of women in the control (60·9%, *p* < 0.001) and CeD groups (67·0%, *p* < 0·001). 

There was a range of gastrointestinal and non-specific symptoms recorded as the predominant symptom at the time of diagnosis (Appendix A). When the structural changes in the histology were classified by Marsh’s grading system [25], 53% of NCGS patients were Marsh 0, and 47% were Marsh I/II. All control participants were Marsh 0 and all in the CeD group were Marsh I-II histology. 

### 3.2. B. Mucosal Characteristics in NCGS, Coeliac, and Normal Mucosa

**B1. Villus Height (VH):** Villus heights were significantly shorter in NCGS patients than in the controls [600 µm (IQR: 400–705) vs. 900 µm (IQR: 667–1112), *p* < 0.001] and significantly longer than the CeD patients [600 µm (IQR: 400–705) vs. 427 µm (IQR: 348–569), *p* < 0.001] (Figure 1 and Appendix A). 

The significance of the shorter villous height in NCGS patients compared to controls persisted, even after limiting the analysis to Marsh 0 specimens [617 µm (IQR: 549–863) vs. 900 µm (IQR: 667–1112), *p* < 0.001]. When comparing villus height in Marsh I/II NCGS with the same grade CeD specimens, NCGS patients had a similar villus height to the CeD group [465 µm (IQR: 390–620) vs. 427 µm (IQR: 348–569, *p* = 0.176). The villus height in Marsh I-II were significantly shorter than controls (p < 0·001), accounting for the country, age, and gender of the patients (Appendix A).

**B2. Crypt Depth (CrD):** The median crypt depth (CrD) was significantly longer (deeper) in the NCGS group than the controls [296 µm (IQR: 205–300) vs. 222 µm (IQR: 158–294), *p* < 0.001) but similar to the CeD group [296 µm (IQR: 205–300) vs. 269 µm (IQR: 182–323), *p* = 0.462] (Figure 2 and Appendix A). 

The longer crypts in NCGS cases compared to controls remained significant even after limiting analysis to Marsh 0 specimens [296 µm (IQR: 261–300) vs. 222 µm (IQR: 158–294), *p* < 0.001]. The CrD values were similar in Marsh I-II for both NCGS and CeD groups [260µm (IQR: 200–296) vs. 269 µm (IQR: 182–323), *p* = 0.409)]. 

**B3. Villus Height to Crypt Depth Ratio (VCR):** The villus to crypt length ratio (VCR) was significantly lower in NCGS patients compared to controls [2.1 (IQR: 1.6–2.5) vs. 4.0 (IQR: 2.9–5.6), *p* < 0.001] (Figure 3, and Appendix A). This was significant even when the analysis was limited to Marsh 0 [2.1 (IQR: 1.8–2.4) vs. 4.0 (IQR: 2.9–5.6), *p* < 0·001]. In contrast, the VCR in NCGS was significantly higher than the CeD group [2·1 (IQR: 1.6–2.5) vs. 1.9 (IQR: 1.3–2.5), *p* = 0.008] in the entire cohort. The VCR was also higher in NCGS than CeD, when the analysis was limited to Marsh I-II [2.1 (IQR: 1.5–2.6) vs. 1·9 (IQR: 1.3–2.5), *p* = 0.019].

### 3.3. C. Intraepithelial Lymphocytes (IELs)

**C1. IELs in Villi:** The median IEL density for NCGS patients [23 (IQR: 17–34)] was significantly higher than controls [14 (IQR: 8–20), *p* < 0.001] and significantly lower than the CeD group [40 (IQR: 31–50) *p* < 0.001] (Figure 4, Appendix A). Marsh 0 NCGS patients had significantly higher IEL densities than controls [18 (IQR: 14–23) vs. 14 (IQR: 8–20), *p* = 0.010]. Comparing IEL densities between the Marsh I/II NGGS and the Marsh I/II CeD group showed slightly lower density in the NCGS mucosa, but the difference did not reach statistical significance [33 (IQR: 27–44) vs. 40 (IQR: 31–50), *p* = 0.224].

To differentiate the NCGS from the control group, the ROC analysis, limited to Marsh 0, revealed 79% sensitivity and 55% specificity at a cut-off of 14/100 enterocytes for NCGS. The area under the curve was 0.70 (95%CI: 0.63–0.76, *p* < 0.001) (Appendix A). IEL densities in Marsh I-II NCGS and CeD groups were similar and the ROC analysis failed to differentiate NCGS from CeD with an area under the curve of 0·43 (95%CI: 0.35–0.65, *p* = 0.065) (Appendix A).

**C2. IELs in Crypts:** The median IEL density in crypts in the NCGS group was similar to the controls [4.0 (IQR: 2.0–7.3) vs. 3.0 (IQR: 2.0–6.0), *p* = 0.121] but lower than the CeD group [6.0 (IQR: 3.0–11.7), *p* = 0.001] (Appendix A). When the comparison was restricted to Marsh 0, NCGS had a crypt IEL density similar to the controls [2.7 (IQR: (1.0–5.0) vs. 3.0 (IQR: 2.0–6.0), *p* = 0.135]. Comparison between crypt IEL densities in Marsh I/II NCGS and the CeD group showed similar results [6.0 (IQR: 2.9–12.0) vs. 6.0 (IQR: 3.0–11.7), *p* = 0.160].

**C3. Percentage of Villi with Increased IEL (PVIIEL):** Data of PVIIEL were available for 550 (79%) cases of the entire cohort. The median PVEIEL in the NCGS group [50% (IQR: 19–89)] was significantly higher than the normal controls [0% (IQR: 0–10), *p* < 0.001] but significantly lower than the CeD group [80% (IQR: 50–100), *p* < 0.001]. When restricted to Marsh 0, PVIIEL in NCGS cases remained higher than the normal controls [20% (IQR: 0–42) vs. 0% (IQR: 0–10), *p* < 0.001]. Comparing PVIIEL in the Marsh I/II NCGS to the Marsh I/II CeD group showed similar percentages in both groups [80% (IQR: 50–100) vs. 80% (IQR: 50–100), *p*= 0.462] (Appendix A).

**C4. Percentage of Villi with even distribution of IEL (PVEIEL):** Data for PVEIEL were available for 513 (73.5%) cases of the entire cohort. The median PVEIEL in the NCGS group was similar to the normal controls [66% (IQR: 38–100) vs. 80% (IQR: 24–100), *p*= 0.631] and the CeD group [66% (IQR: 38–100) vs. 62% (IQR: 40–83), *p*= 0·655]. When the comparison was restricted to Marsh 0, there were no significant differences between PVEIEL in the NCGS and the normal control group [50% (IQR: 26–100) vs. 80% (IQR: 24–100), *p*= 0.637]. Similarly, when the analysis was restricted to Marsh I-II, PVEIEL was similar in NCGS and CeD groups [67% (IQR: 48–90) vs. 62% (IQR: 40–83), *p*= 0·937] (Appendix A).

### 3.4. D. Eosinophils in Lamina Propria

The density of eosinophils in the lamina propria (LPEO) was assessed in 294 (42%) cases. The median LPEO density in NCGS [10 (IQR: 5–15) was similar to the normal controls [15 (IQR: 10–18), *p*= 0.844] but significantly lower than CeD patients [16 (IQR: 7–29), *p* < 0.001]. Patients with NCGS diagnosed by the open challenge approach had similar eosinophil counts to NCGS cases diagnosed by the DBPC method [10 (IQR: 5–19) vs. 10 (IQR: 4–15), respectively *p*= 0.497].

### 3.5. E. Serological Assessment

Quantitative serological data was available for a total of 627 (89.8%) cases of the entire cohort. A positive result on either TTG or EMA measurement was considered a positive CeD serology result. By definition, none of the NCGS patients or normal controls but all CeD patients had a positive TTG or EMA test or positive for tests both. None of the NCGS patients or normal controls were positive but 118 (91.5%) of CeD patients were positive for EMA serology. 

Serum TTG in NCGS was negative and significantly lower than that in CeD patients [(2.0, IQR: 0.6–3.5) vs. (46.0, IQR: 24.0–95.0), *p* < 0.001] but was similar to the normal control group [(2.0, IQR: 0.6–3.5) vs. (1·3, IQR: 0.6–2.4), *p*= 0.289]. Additionally, when we compared TTG titre in Marsh 0 NCGS cases with normal controls (all Marsh 0), results were similar in both groups, being 1.2 (IQR: 0.1–3.1) in NCGS and 1.3 in controls (IQR: 0.6–2.4), *p*= 0.430. When the analysis was restricted to Marsh I and II histological grades, the NCGS group had significantly lower TTG titre than the CeD patients [2.2 (IQR: 1.2–3.7) vs. 46.0 (IQR: 24.0–94.0), *p* = 0.010).

### 3.6. F. Histological Differences of NCGS Patients Diagnosed by DBPC versus OC

Among 175 NCGS patients, 110 (63%) patients were diagnosed by the open challenge technique and the remaining 65 (37%) with the DBPC test. The median ages in the OC and DBPC subgroups were similar [(38, IQR: 30–49) vs. (35, IQR: 25–43), *p* = 0.087]. Female to male ratios were similar in the OC and DBPC subgroups (4.8:1 vs. 4.9:1, *p* = 1.000). Patients in OC subgroup had significantly higher Marsh grades (Marsh above 0) compared to DBPC (58% vs. 30%, *p* = 0.001). Additionally, the OC subgroup had shorter villus heights than patients diagnosed with DBPC tests [500 (IQR: 376–635) vs. 695 (IQR: 600–900) *p* < 0.001. However, their VCR were similar [2.06 (IQR: 1.60–2.51) vs. 2·18 (IQR: 1.80–2.85) *p*= 0.350. The OC subgroup also had higher IEL densities than the DBPC subgroup [27.0 (IQR: 17.0–40.0) vs. 21.0 (IQR: 15.5–30.0) *p* = 0.006, but the two subgroups had a similar eosinophil count, being [10.0 (IQR: 5.0–19.0) vs. 10.0 (IQR: 4.0–15.0), *p*= 0.497], respectively.

## 4. Discussion

This work is a unique global study using conventional histological criteria to elucidate the major mucosal changes associated with non-coeliac gluten sensitivity (NCGS). Our findings show that the intestinal mucosae in NCGS patients exhibited changes consistent with mucosal immune response [4,5,6,26,27] to luminal antigenic stimulation. The NCGS histologic alterations were characterized in 175 patients with the diagnosis, based on the expert Salerno criteria [18], including both double-blind placebo-controlled (DBPC) and open gluten challenge methods. However, no differences were identified between cases diagnosed with the DBPC or open gluten challenge approach. 

Our findings indicate that NCGS mucosae are associated with (i) reduced villus height, (ii) increased crypt depth, (iii) increased lymphocyte infiltration of either villi or crypts and corresponding alterations in villus/crypt ratios even at Marsh 0 stage. On quantification, the TTG titres in NCGS cases were similar to the normal control population but markedly lower than the CeD patients matched for Marsh I/II histology. 

Morphological changes identified in NCGS patients with Marsh 0 challenges the current definition of a normal mucosa, as noted previously [11]. These findings therefore clearly suggest a need for greater precision in delineating normal mucosae from minimal mucosal changes, such as microenteropathy [4]. Our further analysis suggests that intestinal villi in Marsh I-II might be shorter than controls when measured in a morphometric manner, despite their pseudo-normal appearance at microscopy (Appendix A).

A consensus on the extent of enteropathy in NCGS and any potential histological difference between NCGS and CeD is lacking in the current literature. In addition, there are only limited studies [21] that compare the IEL densities and the cut-off between milder enteropathy like Marsh I-II in comparison with Marsh III like pathology. 

We initially believed that IELs are overestimated by a factor of two in patients with severe architectural distortion due to severely damaged epithelial cells [9]. However, a similar IEL cut-off value of 25/100EC for both CeD and NCGS with Marsh I-II lesions in this study brought evidence for the lack of credibility of this point of view. 

In addition, the Marsh 0 stage seems to have a different meaning in NCGS in comparison with normal controls [24], including differences in IEL counts [28] and measurable microstructural abnormalities affecting the villus [29] and crypt architecture (Figure 5). The IELs in the control group mucosae were 13.7/100EC with the majority between 5–10 per 100 epithelial cells. Before the current study, the gap between 5–10 IEL/100EC and the pathological threshold 25 IEL/100EC had not been adequately investigated. Architectural aberrations seem to start from the Marsh 0 stage [25,26,27,28,29] as evident from the significant villus/crypt ratio reduction [30] at this stage in NCGS patients.

Our morphometric measurements as carried out in highly experienced leading tertiary centres indicate progressive remodelling, including changes in villi and crypts, starting from Marsh 0 stage and thus well before the Marsh II stage [29]. The truth seems to be that the expansion of the crypt epithelium and quantifiable changes in villus structure [12,19,31] start as early as the Marsh 0 stage, as demonstrated in our findings (Appendix A). Our study suggests that we need to revisit what can only be described as minimal or subtle alterations, previously defined as “microscopic enteropathies” [4,32,33,34,35]. These alterations must not be dismissed as “non-specific”, as often stated, since these changes nevertheless appear to be a clear response to the stimulating antigen. Instead, they should, perhaps, be regarded as the more subtle changes in the rather wider-scale spectrum of “gluten sensitive enteropathy”, wisely defined by Marsh three decades ago. Nevertheless, despite the progress in our understanding of morphological changes in the small intestinal mucosa, we emphasize that a diagnosis of NCGS based solely on histology would have a poor accuracy or performance, and that there is a need for more specific and sensitive diagnostic markers. Spanish studies report that the intraepithelial lymphocyte (IEL) subpopulations evaluated by flow cytometry are useful to identify a profile named ‘coeliac lymphogram’, consisting of an increase in CD3+ T-cell receptor gamma-delta+ IEL plus a concomitant decrease in CD3- IEL, which is considered highly accurate for CeD irrespective of the degrees of enteropathies [36].

A number of studies recently appeared in the literature concerning lamina propria eosinophils, not only in eosinophilic gastroenteritis [24] but also as a diagnostic aid in NCGS. In one study [20] counts of eosinophils >5 per high-power field (viewed through ×40 objective) in 18 NCGS patients were reported to be a marker of NCGS. In the second study, [22] an increase in eosinophils was also reported, including in rectal mucosa. Therefore, we specifically decided to focus on eosinophils, but as our large number of comparative counts shows, we were unable to verify these findings and the alleged diagnostic utility of this finding reported in these other studies. Indeed, our counts of eosinophils were lower in NCGS patients compared to both the CeD group and the normal controls. Considering the conflicting data from the literature, in addition to the environmental and immunologic factors that can influence the eosinophils in the intestinal mucosa, future studies would be needed to explore the role and cell functional aspects in gluten-induced mucosal inflammation.

***In summary***, based on the validated quantitative histologic analysis [37] and our interobserver agreement (IOA), we identified mucosal alterations associated with NCGS and provided evidence that architectural distortion starts at the Marsh 0 stage. This suggests that the spectrum of what is considered to be normal mucosa needs to be refined further so it can be reliably distinguished from the minimal and subtle alterations of NCGS and CeD. We endorse the IEL cut-off value of 25/100EC in CeD with microenteropathy in addition to demonstrating the existence of mucosal changes that coexist between 5–10 IEL/100EC normal and the pathological threshold of 25 IEL/100EC in NCGS. Lamina propria eosinophils do not appear to be useful for a diagnosis of NCGS.

Finally, our analysis showed no significant histological differences between those patients subjected to a double-blind placebo-controlled diagnostic strategy compared to patients who were included following a simple open challenge approach.

## Figures and Tables

**Figure 1 nutrients-14-02487-f001:**
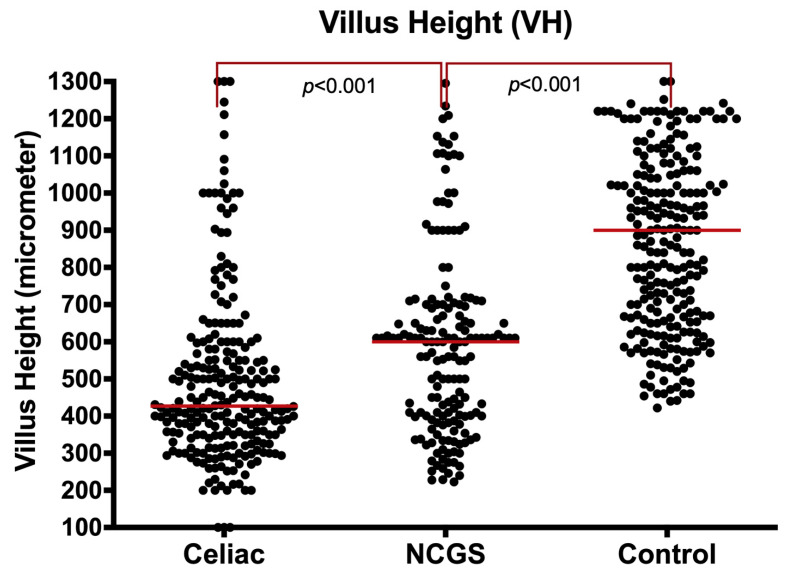
Villus height (μm) in NCGS compared to CeD and controls, entire cohort.

**Figure 2 nutrients-14-02487-f002:**
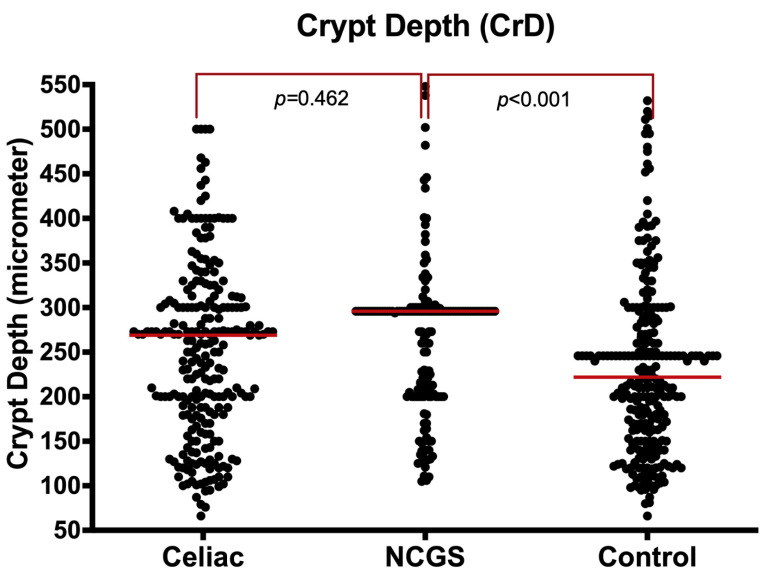
Crypt depth (μm) in NCGS compared to CeD and controls, entire cohort.

**Figure 3 nutrients-14-02487-f003:**
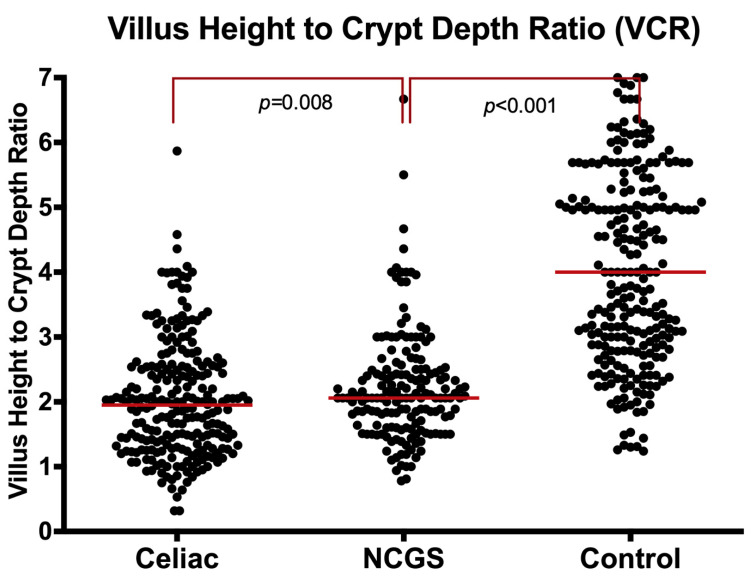
Villus height to crypt depth ratio (VCR) in NCGS compared to CeD and controls, entire cohort.

**Figure 4 nutrients-14-02487-f004:**
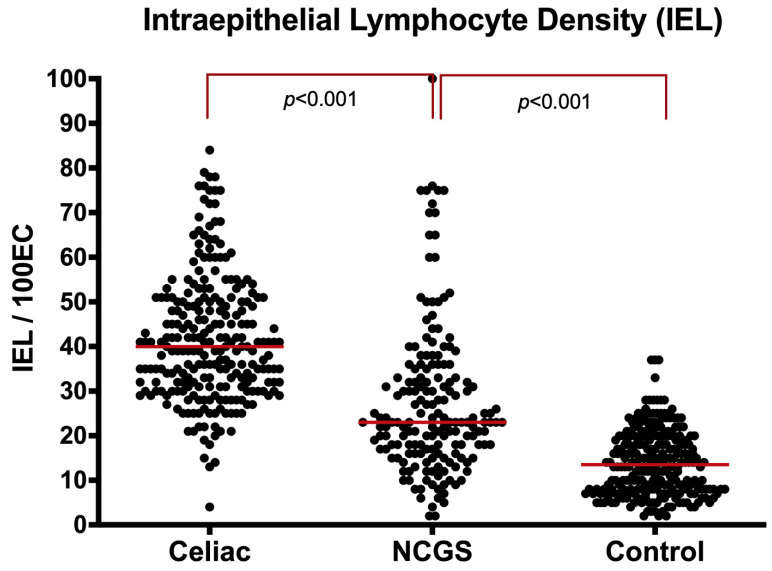
Intraepithelial lymphocytes (IELs) of villi in NCGS compared to CeD and controls, entire cohort.

**Figure 5 nutrients-14-02487-f005:**
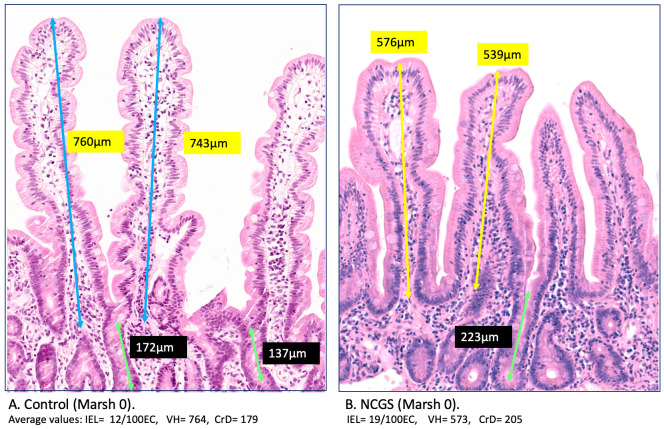
Architectural distortion at Marsh 0 stage. The measurable subtle changes that have been considered as a component of the spectrum of normal mucosa represent a considerable part of architectural distortion signifying the NCGS phenotype (**B**). This reflects in VH, VH/CrD ratio, and the IEL infiltration that were significantly different in NCGS Marsh 0 (**B**) compared to healthy control (**A**).

## Data Availability

Not applicable.

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
