# Peer review of "Gluten Induces Subtle Histological Changes in Duodenal Mucosa of Patients with Non-Coeliac Gluten Sensitivity: A Multicentre Study"

_nutrients, 2022, doi:10.3390/nu14122487_

Round 1
Reviewer 1 Report
Manuscript ID: nutrients-1750724
This is a multicenter and global study to evaluate the mucosal structural and inflammatory features of 175 NCGS (92 Marsh 0 and 83 Marsh I-II) compared to 262 controls (all Marsh 0) and 261 coeliac disease (CD) with milder enteropathy (Marsh I-II). All CD patients had positive celiac serology. Results showed that NCGS duodenal mucosa exhibit distinctive changes consistent with an intestinal response to luminal antigens, even at the Marsh 0 stage of villus architecture.
I have some comments about the study:
1 – Control group is stated that is formed by patients with iron deficient anaemia and weight loss unrelated to upper GI tract with negative celiac serology and normal biopsies (Marsh 0). However, the most frequent symptoms in these patients were abdominal pain and dyspepsia as shown in Table 2 suppl, and only 15% of them had anaemia. This is despite that functional dyspepsia is listed as an exclusion criterion. Thus, this needs to be clarified and described correctly in the methods section. Now, it is not clear how these patients were selected, and it seems that this group would have been selected by a normal histology (IEL count below 25%) and negative celiac serology. This may imply a selection bias.
2- Results show that a percentage of both NCGS (either with Marsh 0 and Marsh I-II) and Marsh I-II CD patients had villous atrophy using morphometry. It would be important to describe this percentage and if there were differences between groups in the percentage of villous atrophy. In addition, it seems pertinent to comment in the discussion if these patients should be catalogued as either seronegative or seropositive CD with atrophy. Also, it should be commented if in the opinion of this group of expert pathologists this methodology should be incorporated in routine clinical practice. It seems cost-effective if a patient can be correctly diagnosed from CD but this should be addressed. In this sense, would it be possible to describe the normal range values for each of these parameters and to recommend which ones and when should be used?
3- The authors give results on anti-tTG titres, but this is not mentioned in the Material and methods section. Based on the inclusion criteria, all controls and NCGS had negative serology whereas CD patients were seropositive. This should be clarified.
4- In the discussion it is stated that there is a need for more specific and sensitive diagnostic markers to differentiate NCGS from CD in patients with ‘microscopic enteropathies’, to correctly diagnose mild forms of celiac enteropathies. In this sense, there is a rising evidence coming from Spain describing that the intraepithelial lymphocyte (IEL) subpopulations evaluated by flow cytometry are useful to identify a profile named ‘celiac lymphogram’, consisting in an increase in CD3+ T-cell receptor gamma-delta+ IEL plus a concomitant decrease in CD3- IEL, which is highly accurate for CD diagnosis even in the seronegative minor forms of CD (p.e., Gut 2002;50:740-1; PLoS One 2014;9:e101249; Nutrients 2019;11(5):1050; Nutrients 2019;11:1992; Aliment Pharmacol Ther 2020;51:699-705; Nutrients 2021; 13:1684; Nutrients 2021;13(6):1812). Results from these studies show that T cell flow cytometry analysis seems to be a promising tool able to differentiate celiac from non-celiac lesions in seronegative patients with high accuracy. A mention of these studies and this methodology in the discussion should be added.
Minor points:
-Table 4 suppl. Define LPI and POVWIIEL (is it a mistake?).
Author Response
Reviewer 1
This is a multicenter and global study to evaluate the mucosal structural and inflammatory features of 175 NCGS (92 Marsh 0 and 83 Marsh I-II) compared to 262 controls (all Marsh 0) and 261 coeliac disease (CD) with milder enteropathy (Marsh I-II). All CD patients had positive celiac serology. Results showed that NCGS duodenal mucosa exhibits distinctive changes consistent with an intestinal response to luminal antigens, even at the Marsh 0 stage of villus architecture.
I have some comments about the study:
1 – Control group is stated that is formed by patients with iron deficient anaemia and weight loss unrelated to the upper GI tract with negative celiac serology and normal biopsies (Marsh 0). However, the most frequent symptoms in these patients were abdominal pain and dyspepsia as shown in Table 2 suppl, and only 15% of them had anaemia. This is despite that functional dyspepsia is listed as an exclusion criterion. Thus, this needs to be clarified and described correctly in the methods section. Now, it is not clear how these patients were selected, and it seems that this group would have been selected by a normal histology (IEL count below 25%) and negative celiac serology. This may imply a selection bias.
Thank you for your comments. The ideal control group would be healthy asymptomatic subjects who are undergoing duodenal biopsies specifically for this study. However the question is how realistic is that approach?. Such a control group requires healthy individuals to undergo endoscopy, sedation and biopsy are very rarely recruited in any studies. There are significant ethical implications that make recruiting a totally healthy group for an invasive investigation and biopsy very unlikely especially in a large global study like this. We acknowledge that our control group was not asymptomatic, however, a diagnosis of CD or NCGS and other upper GI pathology (e.g. H pylori) and functional dyspepsia were excluded before including them in our control group. The endoscopy and histology findings were entirely normal. To address the reviewers concern, we have revised the Methods section and tried to clarify this with more information regarding the control group
2- Results show that a percentage of both NCGS (either with Marsh 0 and Marsh I-II) and Marsh I-II CD patients had villous atrophy using morphometry. It would be important to describe this percentage and if there were differences between groups in the percentage of villous atrophy. In addition, it seems pertinent to comment in the discussion if these patients should be catalogued as either seronegative or seropositive CD with atrophy. Also, it should be commented if in the opinion of this group of expert pathologists this methodology should be incorporated in routine clinical practice. It seems cost-effective if a patient can be correctly diagnosed from CD but this should be addressed. In this sense, would it be possible to describe the normal range values for each of these parameters and to recommend which ones and when should be used?
Thank you. In this study we used multiple morphometric measurements and in doing so, we demonstrate that the villus architecture is not normal in Marsh I-II despite current belief. In addition, we have shown that gluten induces inflammation in NCGS patients even when histology is reported as Marsh 0. This has created an opportunity to characterise the normal mucosa from pathological cases in a more focussed manner. Coeliac and NCGS patients with Marsh I-II show a similar range of inflammation as reflected in our results. In addition, we found that NCGS with Marsh 0 exhibit subtle inflammation when we undertake these morphometric measurements. We rather do not give any other label to this group at this stage, and we hope our finding will be a platform for future investigations that improve our understanding of the diseases in question. Finally, these changes or even more severe architectural distortion like Marsh III are not atrophic changes. According to Marsh these changes are in fact remodelling of small intestinal mucosa into inflammatory hypertrophic changes consisting of an increased crypt epithelial volume surrounding the villus as a collar that ultimately amalgamate and “drown” the villi in the lamina propria leading to a false impression of a flat mucosa. Marsh et al. What Is A Normal Intestinal Mucosa? Gastroenterology 2016;151:784–788, Ensari A, Marsh MN. Exploring the villus. Gastroenterol Hepatol Bed Bench. 2018;11(3):181-190) Though somewhat time-consuming, we encourage using these assessments in routine histology for a more accurate diagnostic value.
3- The authors give results on anti-tTG titres, but this is not mentioned in the Material and methods section. Based on the inclusion criteria, all controls and NCGS had negative serology whereas CD patients were seropositive. This should be clarified.
Many thanks, this information is reported in supplementary material and E section of results Serological Assessment. We would be happy to include the info to material and methods as long as the editors accept that the manuscript word count will be exceeded.
4- In the discussion it is stated that there is a need for more specific and sensitive diagnostic markers to differentiate NCGS from CD in patients with ‘microscopic enteropathies’, to correctly diagnose mild forms of celiac enteropathies. In this sense, there is a rising evidence coming from Spain describing that the intraepithelial lymphocyte (IEL) subpopulations evaluated by flow cytometry are useful to identify a profile named ‘celiac lymphogram’, consisting in an increase in CD3+ T-cell receptor gamma-delta+ IEL plus a concomitant decrease in CD3- IEL, which is highly accurate for CD diagnosis even in the seronegative minor forms of CD (p.e., Gut 2002;50:740-1; PLoS One 2014;9:e101249; Nutrients 2019;11(5):1050; Nutrients 2019;11:1992; Aliment Pharmacol Ther 2020;51:699-705; Nutrients 2021; 13:1684; Nutrients 2021;13(6):1812). Results from these studies show that T cell flow cytometry analysis seems to be a promising tool able to differentiate celiac from non-celiac lesions in seronegative patients with high accuracy. A mention of these studies and this methodology in the discussion should be added.
We thank the reviewer for this insight. We have added a sentence in the Discussion citing the appropriate references:
Spanish studies report that the intraepithelial lymphocyte (IEL) subpopulations evaluated by flow cytometry are useful to identify a profile named ‘celiac lymphogram’, consisting in an increase in CD3+ T-cell receptor gamma-delta+ IEL plus a concomitant decrease in CD3- IEL, considered highly accurate for CD diagnosis even in the seronegative minor forms of CD.
Ruiz-Ramírez P, Carreras G, Fajardo I, Tristán E, Carrasco A, Salvador I, Zabana Y, Andújar X, Ferrer C, Horta D, Loras C, García-Puig R, Fernández-Bañares F, Esteve M. Intraepithelial Lymphocyte Cytometric Pattern Is a Useful Diagnostic Tool for Coeliac Disease Diagnosis Irrespective of Degree of Mucosal Damage and Age-A Validation Cohort. Nutrients. 2021 May 15;13(5):1684
Reviewer 2 Report
I find “Gluten Induces Subtle Histological Changes in Duodenal Mucosa of Patients with Non-coeliac Gluten Sensitivity: A Multi- center Study” a nice and interesting study.
I think supplementary data concerning interobserver agreement methodology should be incoporated to the main document.
I miss information about:
1-The number of anonymized sildes rescored from each group, and the way this number was stated.
2-The number of the 5 expert pathologists evaluating them.
Author Response
I find “Gluten Induces Subtle Histological Changes in Duodenal Mucosa of Patients with Non-coeliac Gluten Sensitivity: A Multi- center Study” a nice and interesting study.
I think supplementary data concerning interobserver agreement methodology should be incoporated to the main document.
I miss information about:
1-The number of anonymized sildes rescored from each group, and the way this number was stated.
In order to validate the quantitative histologic analysis, an inter-observer agreement (IOA) study was planned and a set of 24 anonymized slides comprising well-oriented 8 duodenal biopsies from each group (type 1 CD, NCGS and controls) selected among the original cases included in the study were distributed among the leading expert pathologists (pathology work force comprising AE, VV, AS, PD, GB who are responsible for the design of the histopathological evaluation) and assessed the same histologic morphometric parameters used in the study. An Interclass Correlation Coefficient Analysis was performed on the results to obtain the level of agreement between the observers. Sample size (“number of subjects”) for this analysis (i.e.24 cases) was determined using the two parameters: number of observations per subject and the ICC for both 80% and 90% power levels (see reference below) which ranged from 2 to 15 subjects. The results of Interclass Correlation Coefficient Analysis revealed good to an excellent agreement on IEL counts per 100 enterocytes, villous height (VH), crypt depth (CrD), PVIIEL and PVEIEL, and a moderate agreement was achieved on eosinophil counts in the lamina propria. Data is presented in Supplementary Table 4. (supplementary material)
Bujang MA. A simplified guide to determination of sample size requirements for estimating the value of intraclass correlation coefficient: A review. Arch Orofac Sci (2017), 12(1): 1-11.
2-The number of the 5 expert pathologists evaluating them..
All five expert pathologists (Arzu Ensari (AE), Vincenzo Vilannacci (VV), Amitabh Srivastava (AS), Parsenjit Dass (PD), Gabriel Becheanu (GB) each evaluated all 24 anonymised slides. The sample size was determined according to Bujang et al.
Round 2
Reviewer 1 Report
1. I still have concerns about the description of the control group:
First, in Table 2 suppl, 76 controls had dyspepsia which is considered an exclusion criterion. Therefore, either functional dyspepsia is removed as an exclusion criterion or these 76 controls should be excluded. On the other hand, it is stated that ‘We included patients with anaemia and weight loss unrelated to the upper gastrointestinal tract with normal biopsies’, but in Table 2 suppl only few patients had anaemia. This also should be appropriately described.
Second, I agree that it is difficult to select an adequate control group. In the present study, symptomatic controls have been chosen, many of them with functional GI symptoms, but there is no mention of how NCGS and coeliac disease were excluded. Did they all have negative coeliac serology? Was the response to a gluten-free diet evaluated? If not, this should be recognised as a limitation of the study in the discussion section.
2. I am familiar with the Marsh studies and I would like to reformulate the question I asked in the first review: If a patient classified as Marsh 0-1, after analysis with morphometric techniques is considered to have a Marsh 3 lesion, should he/she be considered to have Marsh 3 seronegative coeliac disease? This aspect is of great importance from an epidemiological, pathophysiological, clinical and treatment point of view and the question is whether the described findings should change routine clinical practice. I believe that all this should be reflected in the discussion, which seems to 'tiptoe' around these aspects.
Author Response
Dear Professor Dr. Maria Luz Fernandez
Thank you for the opportunity to revise our manuscript. We would like to thank the reviewers again for their assessments. This is our rebuttal letter with point-by-point responses by authors. We have adjusted the manuscript following the reviewer’s comment. We hope the manuscript now fulfils the expectations of the reviewers and editors.
Sincerely yours
Kamran Rostami
On behalf of authors
Reviewer 1
- I still have concerns about the description of the control group:
First, in Table 2 suppl, 76 controls had dyspepsia which is considered an exclusion criterion. Therefore, either functional dyspepsia is removed as an exclusion criterion or these 76 controls should be excluded. On the other hand, it is stated that ‘We included patients with anaemia and weight loss unrelated to the upper gastrointestinal tract with normal biopsies’, but in Table 2 suppl only few patients had anaemia. This also should be appropriately described.
Thank you for your comment. We apologise that our use of the term dyspepsia has caused some confusion. To clarify, each centre reviewed the clinical records. We included only “dyspeptic” patients with upper abdominal discomfort associated with diet and or lifestyle. Those with dyspeptic symptoms due to gastric ulcer or H pylori infection were excluded from this study. To prevent confusion, we have changed the term dyspepsia to upper abdominal pain. For the anaemia cases we have the same considerations. For the few cases with a history or clinical manifestation of any disorder or disease, as determined by laboratory finding that was likely to affect the conduct of the study or interpretation of the data, were excluded.
Second, I agree that it is difficult to select an adequate control group. In the present study, symptomatic controls have been chosen, many of them with functional GI symptoms, but there is no mention of how NCGS and coeliac disease were excluded. Did they all have negative coeliac serology? Was the response to a gluten-free diet evaluated? If not, this should be recognised as a limitation of the study in the discussion section.
Many thanks, we acknowledge that the reviewer comment is valid and we have clarified this in manuscript as follows; the control group were included only if they had negative tTG and /or EMA. Most of the control group improved adequately following avoiding the triggering factors related to lifestyle like consuming food rich in sugar/energy/fizzy drinks, fat, high intake of caffeine/alcohol, anxiety/stress or poor mastication habit and those with suboptimal response underwent an elimination diet with gluten without benefit.
- I am familiar with the Marsh studies and I would like to reformulate the question I asked in the first review: If a patient classified as Marsh 0-1, after analysis with morphometric techniques is considered to have a Marsh 3 lesion, should he/she be considered to have Marsh 3 seronegative coeliac disease? This aspect is of great importance from an epidemiological, pathophysiological, clinical and treatment point of view and the question is whether the described findings should change routine clinical practice. I believe that all this should be reflected in the discussion, which seems to 'tiptoe' around these aspects.
Thank you very much for these thoughtful comments.
Please note we are not claiming that these patients with Marsh 0-I have Marsh III lesions. All cases classified as Marsh 0-1 by routine histology showed mild villus/crypt changes on morphometry and none had “flat” mucosa like Marsh III. In fact, we have excluded patients with Marsh III from this study to enable us to focus on the characteristics of NCGS Marsh 0-II and assess how they differentiate from normal mucosa and how they compare with a sub-group of coeliac patients with milder abnormalities classified as Marsh I-II. We agree that seronegative CeD and NCGS have few similarities and we have future plans to assess this aspect in more details in our next studies.
Reviewer 2 Report
I thank the authors for the changes made though I think data about number and selection of slides to be reviewed should be incorpotrated to the main manuscript.
Author Response
Reviewer 2
I think data about number and selection of slides to be reviewed should be incorporated to the main manuscript.
Many thanks
We are happy to add the Interclass Correlation Coefficient information to the main manuscript if the editors allow